# Genome-Wide Analysis and Expression of the *GRAS* Transcription Factor Family in *Avena sativa*

**DOI:** 10.3390/genes14010164

**Published:** 2023-01-06

**Authors:** Lei Ling, Mingjing Li, Naiyu Chen, Guoling Ren, Lina Qu, Hua Yue, Xinyu Wu, Jing Zhao

**Affiliations:** Heilongjiang Provincial Key Laboratory of Oilfield Applied Chemistry and Technology, College of Bioengineering, Daqing Normal University, Daqing 163712, China

**Keywords:** *A. sativa*, *GRAS* transcription factor, genetic structure, abiotic stress

## Abstract

The *GRAS* transcription factor is an important transcription factor in plants. In recent years, more *GRAS* genes have been identified in many plant species. However, the *GRAS* gene family has not yet been studied in *Avena sativa*. We identified 100 members of the *GRAS* gene family in *A. sativa* (*Avena sativa*), named them *AsGRAS1*~*AsGRAS100* according to the positions of 21 chromosomes, and classified them into 9 subfamilies. In this study, the motif and gene structures were also relatively conserved in the same subfamilies. At the same time, we found a great deal related to the stress of cis-acting promoter regulatory elements (MBS, ABRE, and TC-rich repeat elements). qRT-PCR suggested that the *AsGRAS* gene family (*GRAS* gene family in *A. sativa*) can regulate the response to salt, saline–alkali, and cold and freezing abiotic stresses. The current study provides original and detailed information about the *AsGRAS* gene family, which contributes to the functional characterization of *GRAS* proteins in other plants.

## 1. Introduction

*GRAS* transcription factors are important transcription factors in plants, and the *GRAS* gene family is named after GAI, RGA, and SCR [1]. *GRAS* proteins consist of variable N-terminals and the highly-conserved C-terminal. The *GRAS* family members have special *GRAS* domains and high homology at the C-terminal [2]. The C-terminal *GRAS* domain plays an important role in regulating gene expression. The C-terminus includes a few ordered motifs, namely LRI, VHIID, LHRII, LRII, PFYRE, LHRI, and SAW [1].

*GRAS* genes play important roles in regulating the responses to abiotic stresses in plants. Transgenic *BnLAS Arabidopsis thaliana* plants had higher drought tolerances than wild type plants [3]. The *PeSCL7* gene of *Populus* euphratica was transferred into *A. thaliana* (*Arabidopsis thaliana*), which improved the drought resistance and salt tolerance of transgenic plants [4]. Overexpression of *OsGRAS23* increases drought resistance and oxidative stress resistance and reduces hydrogen peroxide accumulation [5]. *VaPAT1* enhances the cold tolerance of grapes by promoting JA biosynthesis [6].

*A. sativa* is an economic grass family of Poaceae. Cultivated *A. sativa* exists as an allohexaploid (AACCDD, 2n  =  6x  =  42), and *A. sativa* has the potential to replace animal-based foods because of its low carbon footprint and great health benefits [7]. Information about the high-quality *A. sativa* allohexaploid genome was published in 2021 [8]. This result also sets the functional characterization of studying the genome-wide analysis of the oat *GRAS* gene family. Oats will be crucial crops for the future and can support crop diversification in response to changing climatic circumstances. Oats are less demanding on soil and they grow well in saline soils; moreover, oats have a strong drought tolerance [9]. Energy expenditure (glycolysis) and biosynthesis (starch and sugar metabolism) were increased in *A. sativa* under abiotic stress [10]. At the same time, the production of *A. sativa* decreased and the average grain weight per plant reduced under abiotic stress [11].

In recent years, more *GRAS* genes have been identified in many plant species, such as alfalfa (*Medicago sativa*), Mignonette (*Melilotus albus*), millet (*Setaria italica*), poplar tree (*Populus*), Tartary buckwheat (*Fagopyrum tataricum*), and pepper (*Capsicum annuum* L.) [12,13,14,15,16,17], but *GRAS* has not yet been studied in *A. sativa*. In this study, we performed a further analysis of the *GRAS* gene family genome in *A. sativa*. We used bioinformatic methods to analyze the gene structures, homologous evolutionary relationships, and gene sequences of *AsGRAS*. The results of our study will help further the investigations into the evolutionary relationships and homologous structures of the *GRAS* gene families and provide a theoretical basis for the functional analysis of *GRAS* proteins under abiotic stress.

## 2. Materials & Methods

### 2.1. AsGRAS Gene Family Identification

We obtained the sequence of members of the *A. sativa GRAS* family from the PlantTFDB database website (http://planttfdb.gao-lab.org/ accessed on 1 November 2022). To find candidate *GRAS* genes, we performed an extensive search and comparison identification for the GRAS domain in *A. sativa* using the blast search and HMMER software (http://hmmer.org accessed on 1 November 2022). We identified the molecular weight (MW), isoelectric point (pI), and length of each GRAS protein through the online ExPASy program (http://www.expasy.org/tools/accessed on 1 November 2022) [18]. Detailed information on the GRAS protein in *A. sativa* is shown in Appendix A.

### 2.2. Analysis of A. sativa GRAS Gene Evolution

The identified *AsGRAS* proteins were combined with the well-sorted *A. thaliana* and *O. sativa* (*Oryza sativa*) *GRAS* proteins and aligned using Clustalx. In addition, we used MEGA7 [19] software to construct the phylogenetic tree. The *AsGRAS* genes were further divided into different subfamilies based on the homology with *GRAS* genes in *A. thaliana* and *O. sativa*. 

### 2.3. Analysis of the Conserved Motif and Gene Structure

To predict the conservative domains of genes, we used the CD-search tool (https://www.ncbi.nlm.nih.gov/cdd/ accessed on 1 November 2022) in NCBI. We extracted exons and introns of the *GRAS* family from the genome annotation information of *A. sativa*, and submitted them to the website tool GSDS for analysis and display. We predicted the matrix of GRAS proteins of *A. sativa* by using the MEME online program [20]. Moreover, we used the TBtools software [21] to comprehensively analyze the conserved motif and gene structure.

### 2.4. Chromosome Localization and Gene Replication Analysis of GRAS Genes

We used Blast software [22] to compare *A. thaliana* and *O. sativa* and selected gene pairs of more than 75% similarity. Then we used Circos software [23] to construct the collinearity plot. Replications of *GRAS* genes were identified and supplemented by using the PGDD database (http://chibba.agtec.uga.edu/duplication/accessed on 1 November 2022) [23].

### 2.5. Cis-Elements Analysis 

To understand the *GRAS* gene family, we analyzed the cis-acting promoter regulatory elements of *GRAS* in *A. sativa*. We detected the sequences within 1500 (bp) upstream of the initiation codons (ATG) for the promoter analysis and looked for these sequences in the oat genome. The cis-elements were searched for through the PlantCARE database in promoters (http://bioinformatics.psb.ugent.be/webtools/plantcare/html/ accessed on 1 November 2022).

### 2.6. Gene Regulatory Network Analysis of the AsGRAS

We used Blast to compare *A. thaliana* and *O. sativa*, selected the protein sequences of the *GRAS* transcription factor in the *A. sativa* genome database, and localized them in the *A. thaliana* Information Resource Database to determine the protein sequences of *A. thaliana GRAS*. We used the PAIR website (http://www.cls.zju.edu.cn/pair/ accessed on 1 November 2022) to predict interactions between *GRAS* and other proteins and a network of interactions in Cytoscape software [24] was exhibited.

### 2.7. Collinearity Analysis of the A. sativa GRAS Gene Families

We used MCScanX technology to look for potential homologous gene pairs (E < 1e^−5^, top 5 matched pairs). By Blast, comparing the gene sequences of *A. sativa* and *A. thaliana* with *O. sativa*, the collinearity between oats and these two species was sought. 

### 2.8. Plant Material and Treatments

*A. sativa* (CV Qinqiong) was used in this study. In the greenhouse, seeds were planted in a 3:1 (*w*/*w*) mixture of soil and sand, germinated, and irrigated with a half-strength Hoagland solution once every 2 days [25]. The seedlings were grown in a night temperature of 18 °C and a day temperature of 24 °C (salt and saline), relative humidity of 60–80%, a 14/10 h photoperiod (daytime, 06:00–20:00), and a light intensity of 200–230 μmol m^−2^ s^−2^. After 4 weeks, the whole germinated seedlings were treated with 150 mM NaCl solution (salt) and 150 mM Na_2_CO_3_ and NaHCO_3_ (saline), at 8 °C (cold), and 4 °C (freeze). Control and treated seedlings were harvested for 6, 12, 24, and 48 h after treatment. Samples were immediately frozen (the whole germinated seedlings) in liquid nitrogen and stored at −80 °C until used for RNA extraction. 

### 2.9. Expression Levels under Stress Treatments

The experimental procedure complied with MIQE guidelines. According to the manufacturer’s protocol, total RNA was extracted from whole seedlings by using RNA to prepare pure plant kits. We stained with a 1.0% (*w*/*v*) agarose gel to assess the RNA quality and removed the DNA contamination with DNaseI treatment. The generated cDNA was stored at −80 °C. Primers were designed by the Primer website (https://www.sangon.com/in_vitro_diagnostic_primers_and_probes.html accessed on 1 November 2022). Moreover, 28 primers were forecasted by the NCBI primer design tool in Appendix A (https://www.ncbi.nlm.nih.gov/tools/primer-blast accessed on 1 November 2022), and 28 *AsGRASs* were quantified under salt, saline–alkali, cold, and freezing abiotic stress; we used SPSS software to analyze differently significant expressions (*p* < 0.05) and the expressions of these genes were analyzed under 4 stresses. 

## 3. Results

### 3.1. AsGRAS Gene Family Identification and Analysis of Physiochemical Properties

We found 100 GRAS proteins in *A. sativa* through the identification and analysis of the *GRAS* gene family (Appendix A). We predicted that among the 100 *GRAS* proteins, the lengths of aminophenol ranged from 160 (*AsGRAS82*) to 1477 aa (*AsGRAS51*), molecular weights ranged from 2406.74 to 7016.67 Da, and isoelectric points ranged from 3.01 (*AsGRAS47*, *AsGRAS57*, and *AsGRAS58*) to 12.08 (*AsGRAS18*).

### 3.2. Analysis of the A. sativa GRAS Gene Family Members and Gene Family Evolution

To study the classification of *GRAS* gene family members, we constructed a phylogenetic tree of *GRAS* protein sequences from *A. thaliana* (37), *O. sativa* (55), and *A. sativa* (100) (Figure 1). Regarding the classification according to the *GRAS* gene family members of *A. thaliana*, the GRAS protein sequences were divided into nine subfamilies (PAT, SHR, SCL, AtSCL3, HAM, GRAS8, DELLA, SCL3, and LAS) of *A. sativa*. Among them, the PAT subfamily had the most members, containing 23 genes and accounting for 23%; the SCL21 subfamily had the fewest members, containing 3 genes and accounting for 3%.

### 3.3. A. sativa GRAS Gene Family Motif Analysis and Gene Structure Analysis

The *AsGRAS* gene family gene structure map (Figure 2) shows that 15 motifs were identified in *GRAS* genes, and motif 6 appeared in every subfamily. Most of the members contained motifs 1–11 except the HAM, GRAS8, and DELLA subfamilies, indicating that this domain is more conserved in this subfamily. The HAM subfamily contains motifs 1–4, 6–10, and 12; the GRAS8 subfamily contains motifs 2–4, 6–8, and 12; and the DELLA subfamily contains motifs 2–4, 6–8, 12, and 14.

*GRAS* gene family members have exon structures, with 34 members without intron structures, accounting for 34%, mainly concentrating in the SHR and PAT subfamilies, and the GRAS8 subfamily has more introns than the remaining subfamily. *AsGRAS18* contains the most introns with a count of 8. The PAT subfamily has no introns, except *AsGRAS13*, *AsGRAS20*, and *AsGRAS19*, and the remaining 20 members have introns that are relatively long. Although most members of the SCL subfamily contain introns, the introns are very short. The GRAS8 subfamily contains introns except for four family members (*AsGRAS25*, *AsGRAS70*, *AsGRAS87*, and *AsGRAS78*) without introns.

### 3.4. Location and Gene Duplication of the Chromosomes of the A. sativa GRAS Gene Family

According to the *AsGRAS* gene duplication map (Figure 3), the *GRAS* genes were randomly identified on each chromosome. We found that 100 *GRAS* genes were mapped to 21 chromosomes. The 12th chromosome had the largest number of gene members on the chromosome, including 10 members, and the highest gene duplication in chromosomes 10, 12, and 20, followed by chromosomes 7, 8, and 9. Among them, there are four gene clusters (Figure 3): *AsGRAS37~43*, *AsGRAS50~57*, *AsGRAS78~81*, and *AsGRAS91~95*.

### 3.5. Analysis of the Promoter Elements of the A. sativa GRAS Gene Family

One hundred *AsGRAS* cis-acting promoter regulatory elements were analyzed, and the elements associated with the stress responses were determined (Appendix A). In this study, four elements were related to the stress responses (MBS, MBSI, ABRE, and TC-rich repeats). Among them, MBS elements and MBSI elements are involved in multiple environmental stresses in plants, ABREs are involved in the abscisic acid response to abiotic stress, and TC-rich repeat elements are involved in the stress response. On the other hand, some other response elements were found, including G-Box photoresponse elements, GARE-motif hormone (gibberellin) response original elements, LTR endogenous retroviral elements, MRE metal response elements, P-Box electronic response elements, and RY-element seed-specific regulatory elements. Thus, it is concluded that the *GRAS* gene can combine with these elements to regulate *A. sativa* growth.

### 3.6. Gene Regulatory Network Analysis of the A. sativa GRAS Gene Family

We constructed an interaction network map of *GRAS* proteins through the relationship between *A. sativa* and *A. thaliana* (Figure 4). Among these proteins, the PAT1 subfamily proteins play vital roles in the early stages of the phytochrome signaling pathway. The SHR and SCR proteins are members of the *GRAS* family of transcription factors and can directly control SCR transcription. The results show that *AsGRAS* transcription factors have complex interaction relationships.

### 3.7. Collinearity Analysis of the A. sativa GRAS Gene Families

By analyzing the collinearity map of *GRAS* genes (Figure 5), we found that the homologous genes of *A. sativa* and *O. sativa* were distributed on 12 chromosomes, and the homologous genes of *A. sativa* and *A. thaliana* were distributed on 5 chromosomes. *AsGRAS* and *OsGRAS* have 85 homologous genes, and *AsGRAS* and *AtGRAS* have 69 homologous genes. In *A. thaliana*, Group A has the highest homology and 26 homologous genes, and Group C has the lowest homology and 26 homologous genes. In *O. sativa*, Group C has the highest homology and 30 homologous genes, and Group A has the lowest homology and 26 homologous genes.

### 3.8. Response and Expression Pattern of A. sativa GRAS Family Genes under 4 Abiotic Stresses (Salt, Saline–Alkali, Cold, and Freezing Stress Conditions)

We selected 28 genes, the expression profiles of which verified their expression patterns under 4 stresses (Figure 6). The results showed that *AsGRAS15*, *AsGRAS17*, *AsGRAS22*, *AsGRAS24*, and *AsGRAS32* expressed the same pattern under salt stress and were upregulated at 12 h. *AsGRAS15* and *AsGRAS17* expressions were upregulated at 48 h, *AsGRAS22* and *AsGRAS32* at 24 h, and *AsGRAS24* at 6 h.

*AsGRAS11*, *AsGRAS12*, *AsGRAS14*, *AsGRAS20*, and *AsGRAS32* were upregulated in early expressions under saline stress, *AsGRAS12*, *AsGRAS20*, and *AsGRAS32* expressions were upregulated at 24 h, *AsGRAS11* at 12 h, and *AsGRAS14* at 48 h.

The results showed that *AsGRAS11*, *AsGRAS14*, and *AsGRAS17* were obviously upregulated at 48 h under cold stress conditions; *AsGRAS16*, *AsGRAS22*, *AsGRAS24*, and *AsGRAS29* were significantly upregulated at 24 h.

Under freezing stress conditions, *AsGRAS14* and *AsGRAS17* expressions were upregulated at 48 h, *AsGRAS15*, *AsGRAS29*, and *AsGRAS32* expressions were upregulated at 12 h, and *AsGRAS16*, *AsGRAS22*, and *AsGRAS32* expressions were significantly upregulated.

## 4. Discussion

We found 100 *AsGRAS* members in this study, and the length of the AsGRAS protein ranged from 2406.74 to 7016.67. The molecular weights of the proteins of tomato, *Medicago truncatula*, and sweet potato ranged from 14,089.3 to 98,029.8, from 7939.17 to 129,425.86, and from 42,500.97 to 157,318.47, respectively [26,27,28]. The results show that the molecular weight of the protein of AsGRAS is less than that of the other plants, and the diversity is lower.

In the *GRAS* gene family, gene duplication is an important method of gene expansion in this family. In this study, the most duplication events were observed in the SCL subfamily of *A. sativa*, the most duplication events were observed in the SCL subfamily of barley and *O. sativa* [29,30]; the most duplication events were observed in the SHR subfamily of *Medicago truncatula* [27]; and the most duplication events were observed in the Pt20 subfamily of the tomato [26]. These results show that duplication events of different subfamilies occupy different main positions in different plants and further verify that gene duplication events in Gramineae were dominated by SCL subfamilies.

In general, the intronless genes are most likely caused by horizontal gene transfer from intronless ancient prokaryotes, duplication of existing intronless genes, or reverse transcription of intron-containing genes [31]. In previous studies, a large proportion of *GRAS* gene families lacked introns. In this study, we found that 34% of *AsGRAS* genes were intronless. A large number of *GRAS* genes are also generally absent in some other plants, such as barley (74.2%), *Medicago truncatula* (88%), tomato (77.4%), and *A. thaliana* (67.6%) [26,27,29,30]. In addition, we found that only 20% of intron genes were involved in duplication events. In the other species, for example, only 4, 6, and 1 intron genes were involved in the duplication events of *A. thaliana*, *O. sativa*, and the tomato [26,30,30]. According to these results, we speculate that genes with introns in the *GRAS* gene family have fewer events involved in gene duplication, and intronless genes are involved in more events in gene duplication, this is because many genes without introns are involved in gene duplication events.

In this study, *AsGRAS11*, *AsGRAS16*, and *AsGRAS17* in the PAT subfamily were significantly upregulated under cold stress, and *AsGRAS11* and *AsGRAS20* were significantly upregulated under salt–alkali stress. Zhang (2018) found that the *Gh_D01G0564* and *Gh_A04G0196* genes in the PAT1 subfamily were significantly increased under salt, drought, cold, and high-temperature stresses in cotton [32]; Yuan (2016) found that the over-expression of the *VaPAT1* gene in *A. thaliana* significantly increased plant resistance to abiotic stress in *Syringa amurensis* [33]. This study shows that the PAT subfamily in the *GRAS* gene family is significantly affected by abiotic stress.

In the present protein interaction regulatory network of the *AsGRAS* gene family, we found that the PAT subfamily proteins interact with proteins such as *SCL21*, and there is an interactive relationship between *AsGRAS16* and *AsGRAS12*. In barley, the PAT1 subfamily proteins (*HvGRAS41* and *HvGRAS50*) may interact with these proteins (*HvGRAS11*) in SCL21 under ethylene reaction, and the PAT1 subfamily members are significantly highly expressed in inflorescence lemma, anatomical inflorescences, inflorescence axis, developmental tillers, and inflorescence [29]. In addition, GRAS proteins can also interact and participate in abiotic stress regulation mechanisms. According to the qRT-PCR analysis in this study, we showed that *AsGRAS16*, a member of the PAT subfamily, can regulate the response to abiotic stresses, such as cold (8 °C) and freezing conditions (4 °C), and *AsGRAS14* can respond to the induction of salinity stress, and its expression is upregulated under salinity stress. In sweet potatoes, the PAT1 protein (*IbGRAS71*) and SHR protein (*IbGRAS4*) are induced under both salt and drought stresses and may participate in stress tolerance or growth and development through complex protein interaction networks [28]. From this result, we conclude that PAT1 subfamily proteins have important functions in the *GRAS* gene family, and multiple stress-responsive *GRAS* genes may play multiple key roles in regulating abiotic stress-signaling cascades through a potentially complex interaction network.

## 5. Conclusions

In this study, we analyzed various bioinformatic data on the *GRAS* evolutionary relationships, gene structure, gene duplication, and promoter analysis of the *GRAS* gene family in *A. sativa*. In this study, through a comprehensive and systematic analysis of the *GRAS* gene family, 100 members of the *GRAS* gene family were named (*AsGRAS1~AsGRAS100*) based on their IDs in the genome. According to the *GRAS* subfamily classification in *A. thaliana*, the *GRAS* gene family is divided into 9 subfamilies, which include most members from the PAT subfamily (containing 23 genes) and the least members from the SCL3 subfamily (containing 3 genes). The *GRAS* genes derived from the same evolutionary clade were found to have similar genes and structures. *GRAS* transcription factors not only participate in the regulation of plant growth and development but also respond to biological stress. The results of this study provide the complete members of the *AsGRAS* transcription factor family and their sequence characteristics, which can provide some reference and theoretical basis for further analysis of the structural characteristics and functional verification of the *AsGRAS* family members.

## Figures and Tables

**Figure 1 genes-14-00164-f001:**
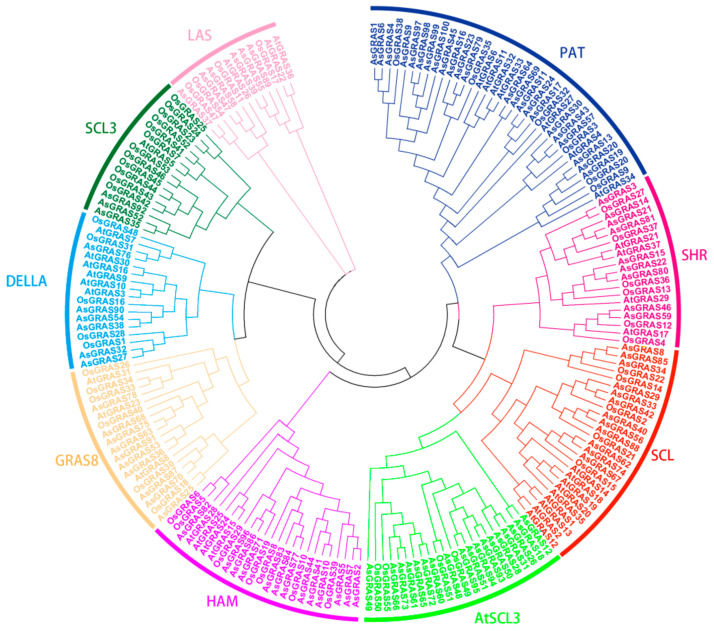
Phylogenetic tree of *Avena sativa* GRAS family members.

**Figure 2 genes-14-00164-f002:**
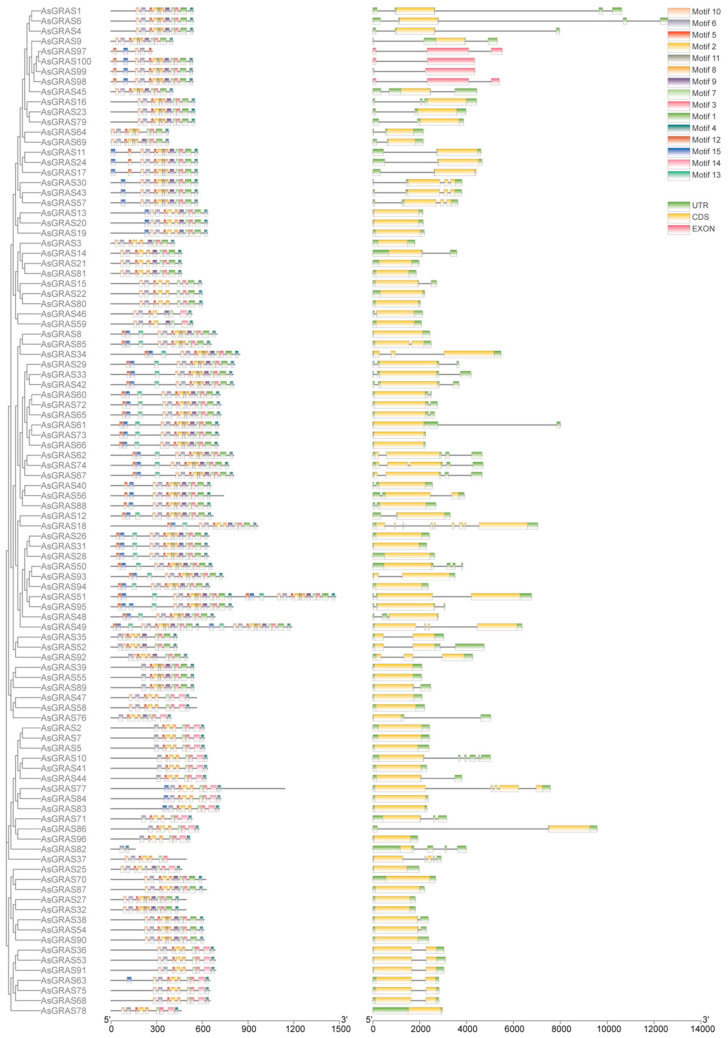
Gene structure analysis of the *Avena sativa* GRAS gene family.

**Figure 3 genes-14-00164-f003:**
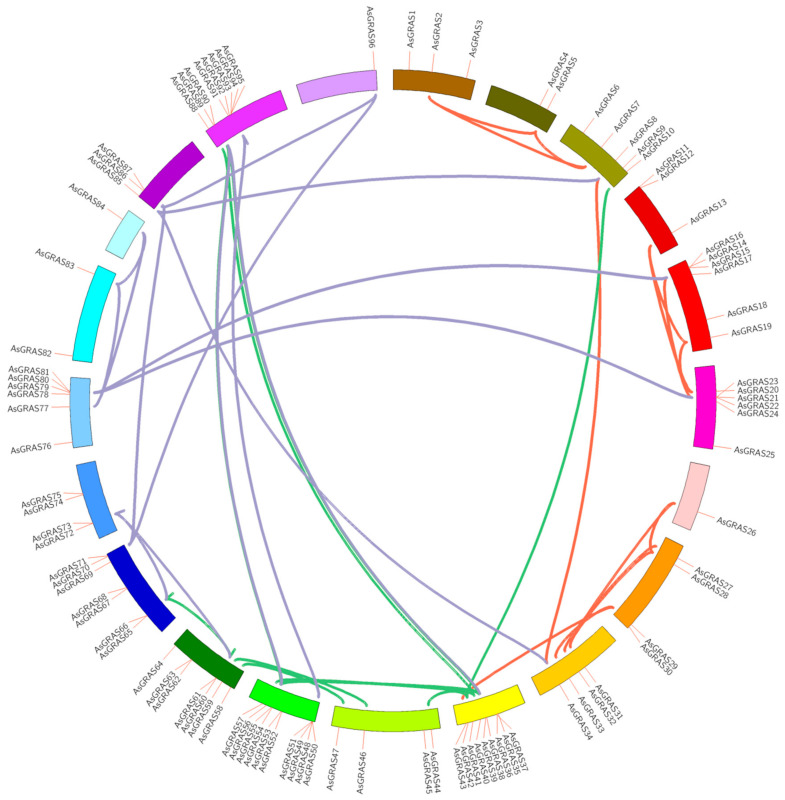
Analysis of the promoter elements of the GRAS gene family in *Avena sativa*.

**Figure 4 genes-14-00164-f004:**
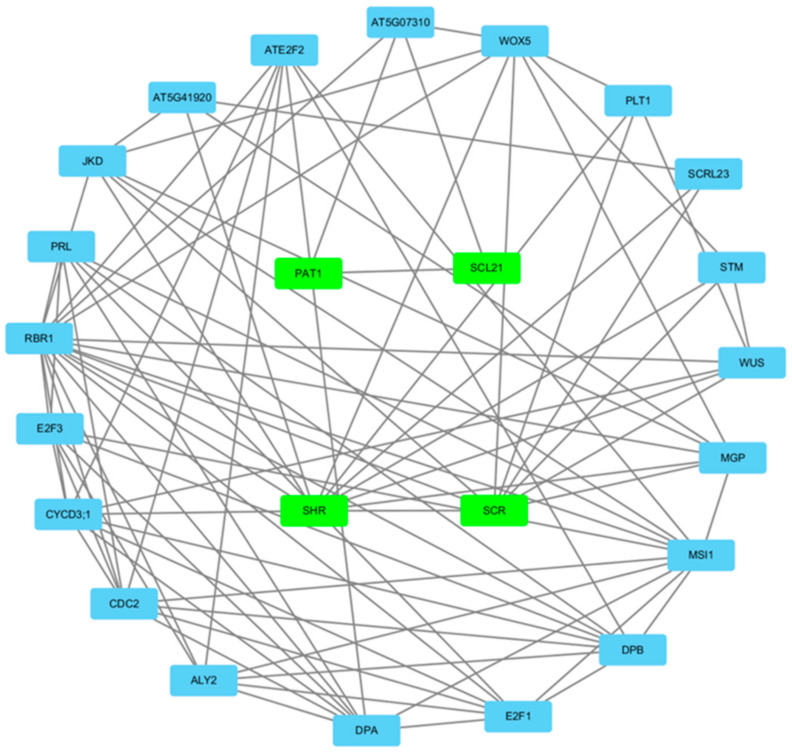
*Avena sativa* GRAS protein interaction network diagram.

**Figure 5 genes-14-00164-f005:**
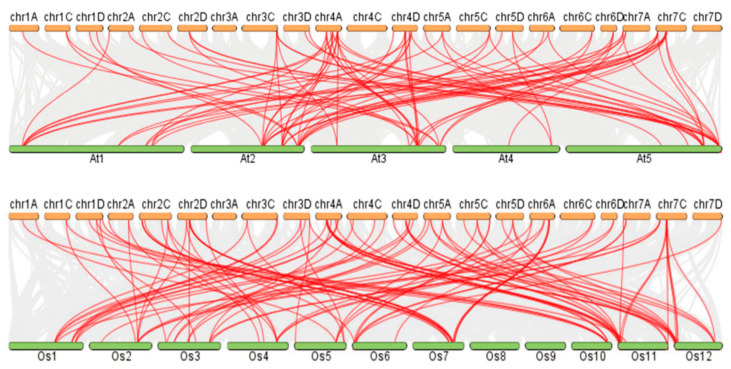
The collinearity map of *GRAS* genes.

**Figure 6 genes-14-00164-f006:**
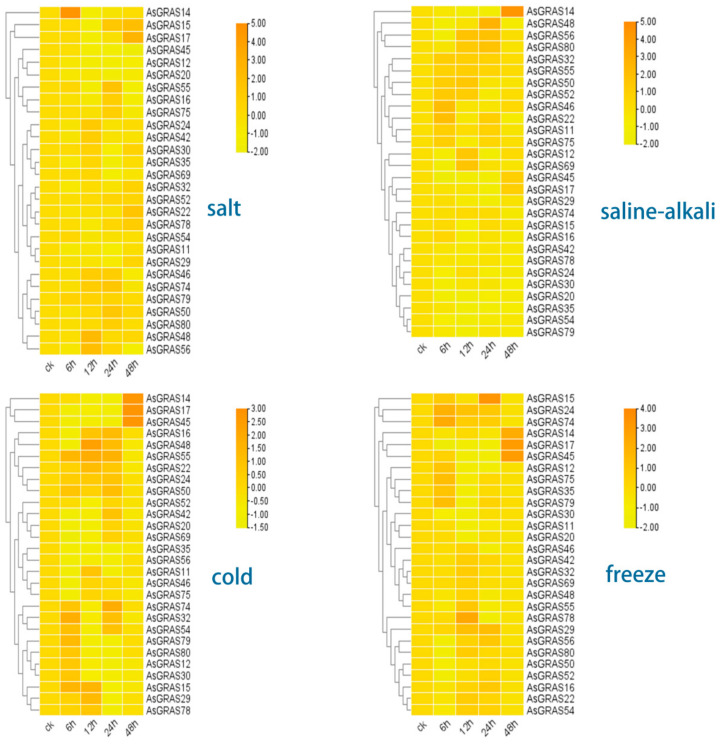
The expression patterns of *GRAS* genes under 4 stresses (salt, saline–alkali, cold, and freezing stress conditions).

## Data Availability

Not applicable.

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
