# Peer review of "Genome-Wide Analysis and Expression of the GRAS Transcription Factor Family in Avena sativa"

_genes, 2023, doi:10.3390/genes14010164_

Round 1

Reviewer 1 Report

GRAS transcription factors (TFs) are essential for both stress tolerance and plant growth and development. Numerous studies have been written about how plants' ability to withstand abiotic stress is influenced by the GRAS transcription factor.

Although GRAS genes have been identified in numerous plant species, there is still some debate on how to classify them. Oats, or Avena sativa, are a crucial crop for the future and can support crop diversification in response to changing climatic circumstances.

For this publication, the authors carried out a genome-wide study and expression of the GRAS transcription factor family in Avena sativa. I

must congratulate the authors for picking the appropriate subject to investigate in terms of the significance of Avena sativa and the GRAS transcription factors because these kinds of studies are desperately needed.

In addition, the study's findings are original and highly fascinating.

Except for a few minor grammatical and typographical errors, the work is quite well written.

However, the author needs to include some more recent references in the introduction chapter.

Overall, the introduction offers adequate context and supports the study's objectives; the conclusion is consistent with the findings of the investigation.

All of the significant issues raised have been covered in the discussion section, and the results are presented in a pleasing manner.

.

Author Response

请参阅附件。

Reviewer 2 Report

the  MS has a lot of grammatical  mistakes, the author need to revise the MS with english native expert.

 the references and citation should be crossed checked and recent citation related to the study should be added. 

figure 1 and 2 are not clearly visible and readable. the pixels and resolution of figures should be improved. 

there should be difference between summary and conclusion. the author just write summary of research rather than conclusion. it is strongly recommended to revise and rewrite the conclusion section. 
